# *FSHβ* links photoperiodic signaling to seasonal reproduction in Japanese quail

Gaurav Majumdar[1], Timothy A Liddle[2], Calum Stewart[2], Christopher J Marshall[2], Maureen Bain[2], Tyler Stevenson[2]*

[1]Department of Zoology, Science Campus, University of Allahabad, Prayagraj, India; [2]School of Biodiversity, One Health and Veterinary Medicine University of Glasgow, Glasgow, United Kingdom

**Abstract** Annual cycles in daylength provide an initial predictive environmental cue that plants and animals use to time seasonal biology. Seasonal changes in photoperiodic information acts to entrain endogenous programs in physiology to optimize an animal's fitness. Attempts to identify the neural and molecular substrates of photoperiodic time measurement in birds have, to date, focused on blunt changes in light exposure during a restricted period of photoinducibility. The objectives of these studies were first to characterize a molecular seasonal clock in Japanese quail and second, to identify the key transcripts involved in endogenously generated interval timing that underlies photosensitivity in birds. We hypothesized that the mediobasal hypothalamus (MBH) provides the neuroendocrine control of photoperiod-induced changes in reproductive physiology, and that the pars distalis of the pituitary gland contains an endogenous internal timer for the short photoperiod-dependent development of reproductive photosensitivity. Here, we report distinct seasonal waveforms of transcript expression in the MBH, and pituitary gland and discovered the patterns were not synchronized across tissues. Follicle-stimulating hormone-β (*FSHβ*) expression increased during the simulated spring equinox, prior to photoinduced increases in prolactin, thyrotropin-stimulating hormone-β, and testicular growth. Diurnal analyses of transcript expression showed sustained elevated levels of *FSHβ* under conditions of the spring equinox, compared to autumnal equinox, short (<12L) and long (>12L) photoperiods. *FSHβ* expression increased in quail held in non-stimulatory short photoperiod, indicative of the initiation of an endogenously programmed interval timer. These data identify that FSHβ establishes a state of photosensitivity for the external coincidence timing of seasonal physiology. The independent regulation of FSHβ expression provides an alternative pathway through which other supplementary environmental cues, such as temperature, can fine tune seasonal reproductive maturation and involution.

**\*For correspondence:**
tyler.stevenson@glasgow.ac.uk

**Competing interest:** The authors declare that no competing interests exist.

## eLife assessment

This **important** article provides insights into the neural centers and hormonal modulations underlying seasonal changes associated with photoperiod-induced life-history states in birds. The physiological and transcriptomic analyses of the mediobasal hypothalamus and pituitary gland offer evidence for a compelling timing mechanism for measuring day length, which is relevant for the field of seasonal biology. The study's **convincing** experiments and findings have the potential to captivate the attention of molecular and organismal endocrinologists and chronobiologists.

## Introduction

Seasonal rhythms in reproduction are ubiquitous in plants and animals. In birds, the annual change in daylength, referred to as photoperiod, provides an initial predictive environmental cue to time

seasonal physiology and behavior (**Ball, 1993**). Temperature, nutrient availability, and social cues act as supplementary cues that function to fine tune the timing of breeding (**Wingfield and Farner, 1980**). Seasonal timing of reproductive physiology and breeding requires the integration of both environmental cues and endogenously generated mechanisms (**Gwinner, 1986**; **Wingfield, 2008**; **Helm and Stevenson, 2014**). Even in the absence of seasonal fluctuations in daylength, temperature, and food availability, endogenous circannual cycles in migration (**Gwinner and Dittami, 1990**), hibernation (**Pengelley and Fisher, 1957**), and reproduction (**Woodfill et al., 1994**; **Lincoln et al., 2006**) are maintained with remarkable temporal precision. Other endogenous timing mechanisms include interval timers that are programmed to establish a physiological state in anticipation of the next season, such as flowering in plants (**Duncan et al., 2015**) and the photorefractory state in rodents (**Prendergast et al., 2001**). The anatomical and cellular basis of endogenous programs that time seasonal transitions in biology remain poorly characterized, but current evidence indicates that, in mammals, circannual time may reside in pituitary lactotropes (**Lincoln et al., 2006**) and thyrotropes (**Wood et al., 2020**).

In most long-lived species (e.g., >2 years), the annual change in photoperiod acts to entrain endogenous annual programs to time transitions in physiological state to seasons (**Bradshaw and Holzapfel, 2007**). In many temperate-zone birds, exposure to long days (>12 hr), induces a photostimulated state in which gonadal development occurs in male and female birds. Short photoperiods (e.g., <12 hr) can induce gonadal involution in both sexes leading to a reproductively regressed state (**Dawson et al., 2001**). Birds become reproductively sensitive to stimulatory long photoperiods only after experiencing short photoperiods for at least 10 days, in which a photosensitive state is established (**Dawson et al., 2001**).

In most birds, reptiles, and amphibians, annual changes in daylength are detected by photoreceptors located in the mediobasal hypothalamus (MBH) (**Pérez et al., 2019**). Stimulatory long photoperiods trigger a molecular cascade that starts with the upregulation of thyrotropin-stimulating hormone-β (*TSHβ*) subunit in the pars tuberalis of the pituitary gland and results in gonadal maturation (**Nakao et al., 2008**). The current gap in our knowledge is how short days (i.e., <12 hr) induce gonadal involution, and how prolonged exposure to short days stimulates endogenous programs that sensitizes the brain to respond, at a molecular level, to stimulatory long days (**Follett and Sharp, 1969**). Previous studies in European starlings (*Sturnus vulgaris*) demonstrated that exposure to short days increased gonadotropin-releasing hormone (*GNRH*) expression in the preoptic area (**Stevenson et al., 2009**; **Stevenson et al., 2012a**). As both GnRH and gonadal growth increased in the absence of stimulatory long daylengths, another cellular pathway must also be involved in the endogenous development of reproductive physiology in birds.

There were three objectives of the present work. First, we aimed to characterize the photoperiod-induced seasonal molecular clock in the MBH and pituitary gland in Japanese quail. Then, we examined the daily waveform of multiple transcripts in the MBH and pituitary in birds from stimulatory long photoperiod (16L:8D), inhibitory short photoperiod (8L:16D), and the two equinoxes. The last objective was to determine if follicle-stimulating hormone-β (*FSHβ*) expression in the pituitary gland was upregulated after prolonged exposure to short photoperiods. We theorized multiple neuroendocrine regions including the preoptic area, MBH, and pituitary cells are independently involved in the timing of seasonal transitions in physiology. Specifically, we hypothesized that pituitary gonadotropes establish a state of photosensitivity to stimulatory long day photoperiods and thus act as calendar cells that provide endogenous timing of gonadal growth.

## Results

### Molecular characterization of the photoperiod-induced seasonal clock

To obtain a comprehensive understanding of the seasonal molecular changes in the MBH and pituitary gland, we collected MBH and pituitary gland samples from Japanese quail using an experimental paradigm that aimed to maximize resolution (i.e., high sampling frequency), high dimensionality (i.e., advanced nucleic acid sequencing), and robust statistical power (i.e., large sample sizes). Our experimental design simulated the photoperiodic regulation of seasonal physiology of Japanese quail, using sequential changes of an autumnal decrease, followed by a spring increase in daylength and measured testes volume, body mass, and abdominal fat (**Figure 1**). As anticipated (**Follett and Sharp, 1969**; **Robinson and Follett, 1982**), light phases that exceed the critical daylength (i.e.,

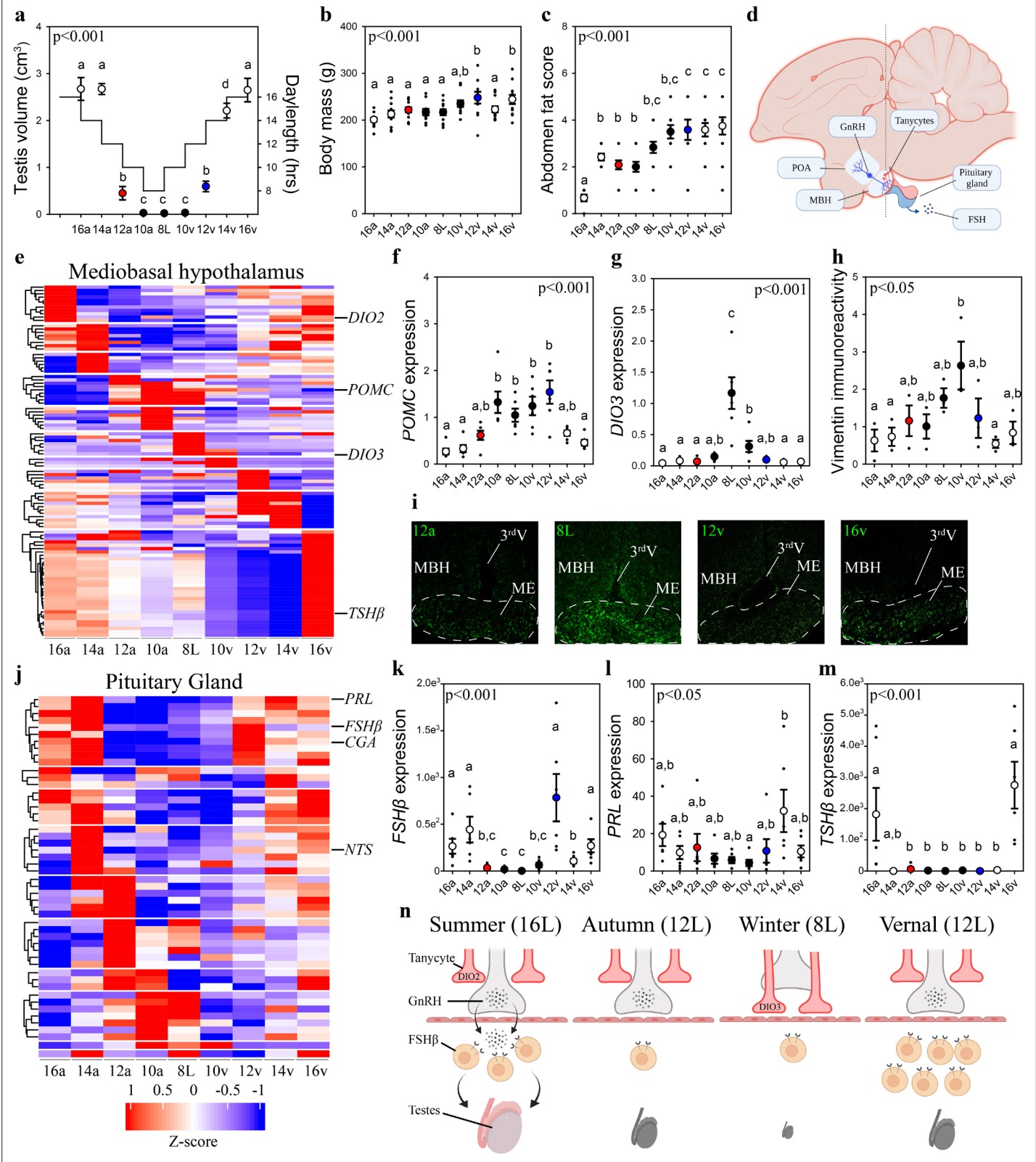

**Figure 1.** Vernal increase in pituitary *FSHβ* expression precedes the molecular switches in mediobasal hypothalamus and gonadal growth. (**a**) Schematic representation of the simulated annual rhythm in photoperiod. Quail were collected in 16 hr light, 8 hr dark photoperiod and then every 2 weeks the photoperiod was decreased by 2 hr to 14 hr, 12 hr, 10 hr, and then an 8L short photoperiod. Photoperiod was then increased to mimic the vernal transition and birds were collected at 10, 12, 14, and 16 hr light photoperiods. Testis volume confirmed critical daylength (i.e., 12 hr) induced

*Figure 1 continued*

growth. (**b**) Body mass and (**c**) abdominal fat deposition increased until the autumnal equinox (12a), and then increased during the vernal photoperiod transitions. (**d**) Diagram highlighting hypothalamic preoptic area (POA), mediobasal hypothalamus (MBH), and pituitary gland. Tanycytes in the MBH gate GnRH release into the pituitary. (**e**) Heatmap of RNA-seq of MBH punches identified distinct wave of transcripts as quail transition across photoperiodic conditions. (**f, g**) Quantitative PCR (qPCR) assays for proopiomelanocortin (*POMC*) and deiodinase type-3 (*DIO3*) confirmed restricted activation during 10a–8L and 8L–10v phases, respectively. (**h, i**) Vimentin immunoreactivity in the median eminence (ME) show tanycytes morphology growth is limited to 10a, 8L, and 10v photoperiods. (**j**) Heatmap illustrating photoperiodic transitions in pituitary transcripts. (**k–m**) qPCRs confirmed that follicle-stimulating hormone-β (*FSHβ*) is elevated under non-stimulatory photoperiods followed by increased prolactin (*PRL*) in 14v and thyrotropin-stimulating hormone-β (*TSHβ*) in 16v. (**n**) Diagram summarizing that long photoperiods increased *GNRH* synthesis and release into the pituitary gland to stimulate *FSHβ* and induce testis growth. Transition to autumnal equinox phases results in reduced *FSHβ* expression and regressed testis. Prolonged exposure to short photoperiods inhibits *GNRH* expression, triggers tanycyte extension, maintains low *FSHβ*, and regressed testis. Vernal transitions in photoperiod to the equinox results in resumption of *GNRH* and elevated *FSHβ* expression without testis growth. Data are mean ± standard error of the mean (SEM), and residual dot plot. (**a–c, f, g, k, m**) One-way analysis of variance (ANOVA) with Bonferroni corrected Tukey's test for multiple comparisons. (**h, l**) One-way ANOVA with Tukey tests for significant pairwise comparison. Letters denote significant difference between photoperiod phases. Raw data available in *Figure 1—source data 1*.

The online version of this article includes the following source data and figure supplement(s) for figure 1:

**Source data 1.** Raw data.

**Source data 2.** Seasonal MBH gene ontology analyses.

**Source data 3.** Biodare2.0 analyses of seasonal mediobasal hypothalamus (MBH) transcripts.

**Source data 4.** Seasonal pituitary gland gene ontology analyses.

**Source data 5.** Biodare2.0 analyses of seasonal pituitary gland transcripts.

**Source data 6.** Weighted gene co-expression network analyses.

**Source data 7.** Transcription-binding motifs in the mediobasal hypothalamus transcripts.

**Source data 8.** Transcription-binding motifs in the pituitary gland transcripts.

**Figure supplement 1.** Heat-map of significant transcripts determined by BioDare2.0 for the mediobasal hypothalamus and pituitary gland.

**Figure supplement 2.** Assessment of photoperiodic, epigenetic and transcription-binding factors in the mediobasal hypothalamus and pituitary gland.

**Figure supplement 3.** Weighted gene co-expression network analyses of the mediobasal hypothalamus (MBH) and pituitary gland transcripts.

**Figure supplement 4.** CHiP-X enrichment analyses to identify DNA-binding motifs common across transcripts detected as significantly different based on BioDare2.0 analyses of the mediobasal hypothalamus (MBH) (**a**) and pituitary gland (**b**).

>12 hr) resulted in robust gonadal growth. Increases in body mass and abdominal fat deposition were delayed until the spring increase in daylengths reached 10 hr (10v) (*Figure 1*). EdgeR analyses, of MBH sequences obtained using Minion, identified 1481 transcripts were differentially expressed ($p < 0.05$) (*Figure 1—figure supplement 1*; *Figure 1—source data 1 and 2*), and BioDare2.0 established 398 have rhythmic patterns (*Figure 1—source data 1–3*). DAVID gene ontology analyses indicated that gonadotropin-releasing hormone receptor and Wnt signaling pathways were consistently identified as the predominant cellular mechanism recruited during each photoperiodic transition. Increased proopiomelanocortin (*POMC*) expression coincided with body mass growth and abdominal fat accumulation (*Figure 1*). Thyroid hormone catabolism enzyme deiodinase type-3 (*DIO3*) increased after prolonged exposure to non-stimulatory photoperiods and was only therefore transiently elevated from 8L to 10v (10v; *Figure 1*).

Next, we used vimentin immunoreactivity to examine changes in tanycytes morphology in relation to changes in deiodinase transcript expression. Highest levels of vimentin immunoreactivity in the median eminence coincided with the peak in *DIO3* expression (*Figure 1*) suggesting the localized removal of active thyroid hormone is limited to a short phase and occurred prior to stimulatory photoperiods (*Figure 1*). The photoperiod-induced change in vimentin was anatomically localized to the median eminence, as the area of immunoreactivity in the dorsal 3rdV ependymal layer did not change with seasonal transitions in reproduction (*Figure 1—figure supplement 2*).

Next, we used MinION to sequence transcripts in the pituitary gland across the seasonal transitions in reproduction. EdgeR analyses identified 3090 transcripts were differentially expressed in the pituitary gland (*Figure 1—source data 1*). DAVID gene ontology analyses identified that gonadotropin-releasing hormone receptor and epidermal growth factor pathways are consistently observed across photoperiodic transitions (*Figure 1—source data 4*). BioDare2.0 established that 130 transcripts were rhythmically expressed (*Figure 1—source data 5*). A remarkable 96.5% (384/398) of MBH transcripts

show a photoperiod-induced spiked patterned and only 12/14 remaining transcripts displayed sine waveforms. Conversely, 100% of pituitary transcripts conformed to sine (124/130) or cosine (6/130) waveforms. The predominant rhythmic patterns of expression likely reflect endogenous long-term molecular programs, characteristic of calendar cell function.

Lastly, we examined *GNRH* expression to delineate photoperiod-induced changes in the neuro-endocrine control of reproductive physiology. *GNRH* expression in the preoptic area was temporarily decreased during the 8L short photoperiod indicating another nucleus-specific cellular timer is present in this brain region (*Figure 1—figure supplement 2*). Overall, the pituitary showed a distinct tran-scriptomic profile compared to the MBH suggesting independence in the representation of seasonal photoperiodic timing. The spring photoperiodic transition from 8L to 12v resulted in a significant increase in *FSHβ* without a change in tanycyte restructuring (*Figure 1*). Subsequent transition to 14v and then 16v resulted in the upregulation of prolactin (*PRL*) and *TSHβ*, respectively. These data estab-lish that multiple cells in the pituitary code seasonal photoperiodic time and *FSHβ* shows an endoge-nous increase in expression prior to reproductively stimulatory photoperiods.

Weighted gene co-expression network analyses (WGCNA) were conducted to discover gene co-expression modules, and then examine whether any of the resulting module eigengenes co-vary with photoperiod or physiological measures. The eigengene dendrogram of sequencing from indi-vidual animals was plotted and a heatmap of physiological factors was organized (*Figure 1—figure supplement 3*). The scale-free topology and mean sample independence was assessed to determine a soft-threshold of 5 for both MBH and the pituitary gland sequencing datasets. Ten modules were identified for the MBH, and the pituitary gene set was grouped into 22 modules. Of these modules there were 6 significant module–trait relationships in the MBH. There was one module with a signifi-cant negative correlation with fat score (*Figure 1—figure supplement 3*; *Figure 1—source data 6*). Forty-four transcripts were identified to be significant in the negative relation for fat score. The other five were identified to be negatively related and included photoperiod, body mass, fat score, testes width, and testes volume (*Figure 1—figure supplement 3*; *Figure 1—source data 6*). Overall, there were 23 transcripts that were significant and overlapped with photoperiod, testes width, and testes volume. The other module found 70 transcripts for both body mass and fat score. Despite several modules showing trends toward significance, only one module for body mass was positively related in the pituitary gland (*Figure 1—figure supplement 3*). There were 206 transcripts identified to be significantly positively related to body mass.

To ascertain common molecular mechanisms involved in the transcriptional regulation of photope-riodically regulated transcripts, transcription factor enrichment analysis was conducted on significant MBH (*Figure 1—source data 7*) and pituitary gland (*Figure 1—source data 8*) transcripts. Associa-tion plots show no overlap in DNA-binding motifs between MBH and pituitary transcripts (*Figure 1—figure supplement 4*) suggesting tissue-specific transcription-binding factor regulation. Within the pituitary gland, several common transcription factors, such as the Jun proto-oncogene, Spi1 proto-oncogene, and myocyte enhancer factor 2 (*MEF2a*) might be actively involved in the photoperiodic regulation of transcript expression. These findings indicate multiple transcription factors are likely recruited to control tissue-specific, and cell-specific transcript expression and seasonal life-history transitions in physiology.

## Increased *FSHβ* expression is programmed during the spring equinox

To establish whether increased pituitary *FSHβ* during the spring 12 hr transition reflects constitu-tively elevated expression or is driven by the sampling 'time of day', we collected tissue samples every 3 hr from quail that transitioned to the autumnal equinox (i.e., 12a), the short photoperiod (8L), spring equinox (12v), and long photoperiod (16L) seasonal phases. Photoinduced changes in testes volume confirmed seasonal reproductive condition (*Figure 2*, *Figure 1—source data 1*). Circa-dian clock genes aryl hydrocarbon receptor nuclear translocator like (*ARNTL1*) and period 3 (*PER3*) exhibit robust anti-phase daily waveforms in expression, in the pituitary gland (*Figure 2—source data 1*). Only *ARNTL1*, but not *PER3* nor *DIO2*, had a rhythmic waveform in the MBH (*Figure 2—figure supplement 1*; *Figure 2—source data 1*). Consistent with the previous study, *FSHβ* expression was higher at the spring equinox compared to all other photoperiod groups (*Figure 1—source data 1*). *PRL* expression was higher in long photoperiod (16L) compared to the two equinox and short photo-period (8) (*Figure 1—source data 1*). *FSHβ* did not display a daily rhythm, which is likely due to the

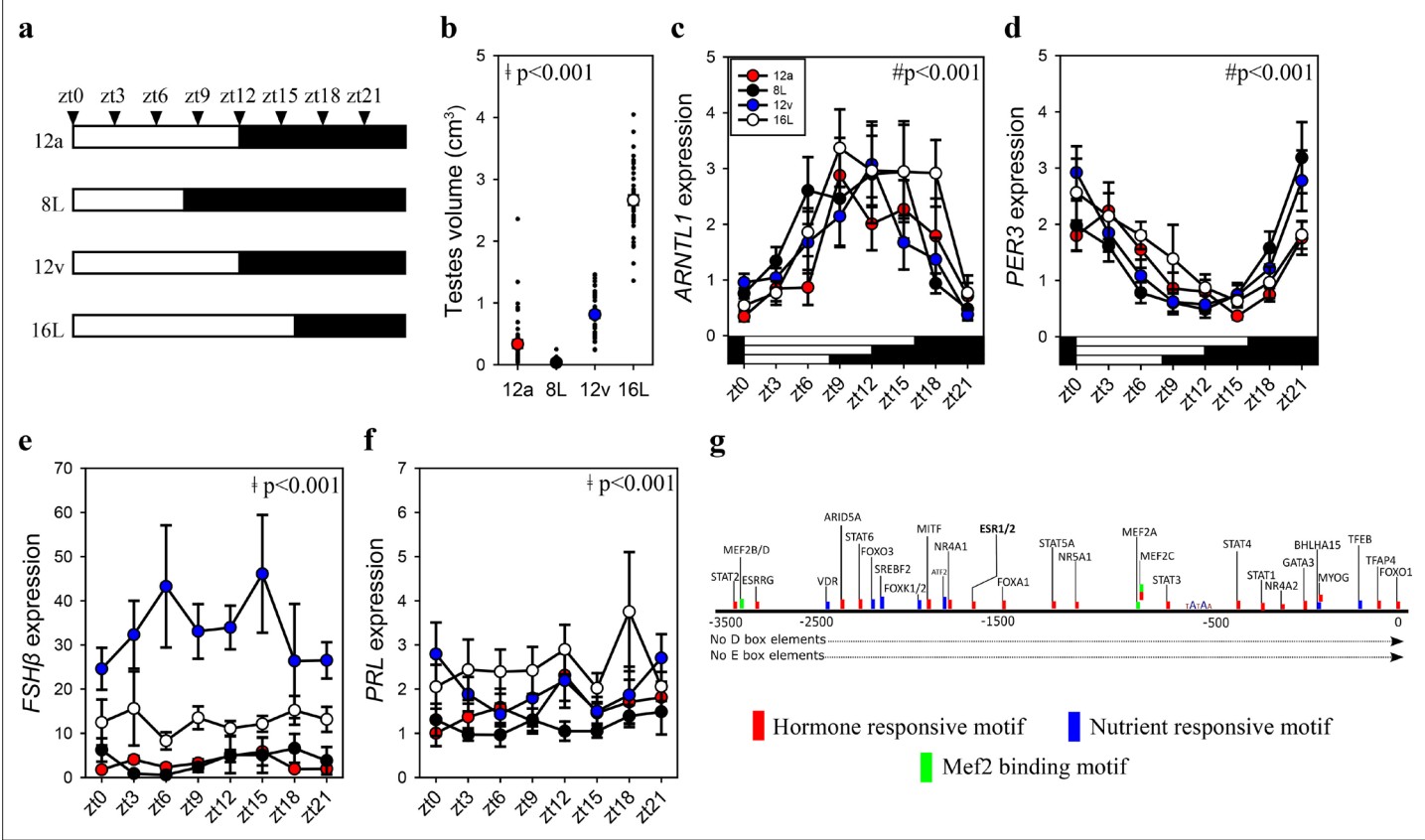

**Figure 2.** *FSHβ* expression is constitutively expressed during the vernal equinox. (**a**) Schematic representation of four photoperiod treatment groups with arrows to indicate the daily sampling time. (**b**) Testes volume remained in a regressed non-functional state in autumnal equinox (12a), short photoperiod (8L), and vernal equinox (12v). Photoperiods that exceeded the critical daylength (i.e., >12 hr) induced testes growth. (**c**) Pituitary circadian clock gene *ARNTL1* maintained daily rhythmic expression waveforms across all photoperiods (p < 0.001), there were no significant differences between photoperiod treatments (p = 0.42). (**d**) *PER3* displayed a daily waveform across 12a, 8L, 12v, and 16L groups (p < 0.001) and was anti-phase compared to *ARNTL1*. There was no significant difference between photoperiod treatment (p = 0.31). (**e**) Follicle-stimulating hormone-β (*FSHβ*) expression was significantly higher in 12v compared to long photoperiod (16L; p < 0.001), autumnal equinox (12a; p < 0.001) and short photoperiod (8L; p < 0.001) but was not rhythmic (p = 0.66). (**f**) Similarly, *PRL* was high in 16L compared to 12a (p < 0.001), and 8L (p <0 .001), there was no significant daily rhythms (p = 0.52). (**g**) *FSHβ* promoter was devoid of circadian gene-binding D- and E-box motifs but contains a series of hormone and nutrient responsive motifs. (**b–f**) Two-way analysis of variance (ANOVA) followed by Tukey's pairwise tests, rhythmic analyses were conducted using GraphPad Prism. ‡ indicates significant photoperiod treatment effects; # denotes significant time of day effect. Data are mean ± standard error of the mean (SEM) and residual dot plot (**a–d**). Raw data are available in *Figure 1—source data 1*.

The online version of this article includes the following source data and figure supplement(s) for figure 2:

**Source data 1.** Daily waveform analyses.

**Source data 2.** Follicle-stimulating hormone-β (*FSHβ*) promoter DNA motif and transcription-binding factors.

**Figure supplement 1.** Circadian clock gene (*ARNTL1*, *PER3*) in the mediobasal hypothalamus (MBH) from Japanese quail collected every 3 hr under short photoperiod (8L), autumnal equinox (12a; 12L:12D), vernal equinox (12v; 12L:12D), and long photoperiod (16L).

absence of D- and E-box motifs in the *FSHβ* promoter (**Figure 2**). *FSHβ* promoter does contain many DNA motifs that are targeted by several transcription factors that are responsive to hormonal and nutrient pathways indicating multiple upstream regulators are recruited to drive transcription (**Figure 2—source data 2**). These data support the conjecture that a long-term programmed increase in *FSHβ* occurs under spring non-stimulatory photoperiod, and it is not driven by short-term daily photic cues.

## *FSHβ* expression establishes endogenously programmed photosensitivity

To identify if *FSHβ* expression is driven by an endogenously programmed mechanism or in response to the gradual increase in light, adult quail were exposed to the 8L, 10v, or 12v light schedules or kept in

8L for an additional 4 weeks (8Lext) (*Figure 3—figure supplement 1*, *Figure 3*). The 8Lext treatment permitted confirmation whether *FSHβ* expression would increase in that photoperiod, and therefore reflect an interval timing mechanism. *FSHβ* expression increased 19-fold after four additional weeks of 8L suggesting that endogenous drivers initiate transcription despite no change in daylength (p < 0.05) (*Figure 1—source data 1*). But the dominant stimulator of *FSHβ* expression was the transition to 10v and 12v photoperiod, which both expressed significantly increased levels compared to 8L (p < 0.001). These data demonstrate that both photoperiod and endogenous timing mechanisms drive *FSHβ* expression in the pars distalis. It is likely that an additive function of endogenous timing and the spring increase in photoperiod drive *FSHβ* expression.

As Opn5 was identified in the pituitary transcriptome, we then assessed its transcript expression across the photoperiod treatments. We discovered that Opn5 expression patterns paralleled *FSHβ* suggesting the potential for direct light detection by Opn5 and subsequent regulation of *FSHβ* expression. Based on *FSHβ* promoter analyses (*Figure 2*), we examined the expression of myocyte enhancer factor 2 (*MEF2a*) expression as a potential upstream regulator of *FSHβ* expression. *MEF2a* remained relatively constant suggesting this transcription factor-binding protein is not the primary driver of *FSHβ* expression (*Figure 3—figure supplement 1*). Similarly, *DNMT3a* expression did not change across photoperiod treatments (*Figure 3—figure supplement 1*) suggesting that epigenetic modifications (i.e., DNA methylation) may not provide the endogenous programmed change leading to constitutive *FSHβ* expression. The precise molecular change upstream from *FSHβ* transcription remains to be identified. In the MBH, *DIO3* displayed a rapid reduction in expression after transfer to 10v and was found to be significantly reduced in 12v photoperiod (p < 0.001) (*Figure 1—source data 1*). There was a significant difference in MBH *DIO2* expression (*Figure 3—figure supplement 1*), but this observation was driven by a decrease in 8ext. Interestingly, *GNRH* in the preoptic area was found to significantly increase after extended exposure to short photoperiods (*Figure 1—source data 1*). These data indicate that endogenous switches in *FSHβ*, and possibly *GNRH* expression, in response to short photoperiods may reflect multiple independent cellular timers that establish a physiological state of photosensitivity.

## Discussion

This report used the well-characterized photoperiodic manipulation of the Japanese quail avian photoperiodic response using a laboratory-based light schedule that accurately replicated findings from birds held in semi-natural conditions (*Robinson and Follett, 1982*). The data reported herein demonstrate that photoperiods less than 12 hr light induce gonadal involution. Prolonged exposure to short photoperiods (i.e., 8) were found to significantly increase *DIO3* expression and vimentin immunoreactivity in the median eminence. Increased *DIO3* expression and innervation of tanycytes occurred after gonadal regression suggesting that another unidentified mechanism is involved in the initiation of the termination in the breeding state. The gradual increase in photoperiods during the spring transition was found to be associated with a marked increase in *FSHβ* expression in the pars distalis while lower levels of *TSHβ* expression in the pars tuberalis were maintained. As photoperiods increase there is a steady elevation in *FSHβ* expression, but vimentin immunoreactivity in the median eminence did not decline until after the critical daylength for photostimulation (i.e., 12L:12D) thus the release of FSH is prevented as daylengths are below the critical threshold. Previous reports established that *TSHβ* expression is significantly increased during the period of photoinducibility in quail (*Nakao et al., 2008*). Although the present study did not directly examine photoinduction, *TSHβ* expression was consistently elevated in long day photoperiod (i.e., 16L). The patterns of expression suggest that stimulatory daylengths longer than 12 hr induce thyrotropes to increase *TSHβ* leading to a cascade of molecular events in the MBH that permit GnRH to stimulate gonadotropes to release FSHβ and initiate gonadal development (*Nakao et al., 2008*; *Yoshimura et al., 2003*; *Yamamura et al., 2004*). Note that other pituitary cell types, somatotropes and corticotropes do not appear to show any molecular switches across the photoperiodic phases. These findings uncover a two-component mechanism for the cellular basis of the external coincidence model for the avian photoperiodic response (*Supplementary file 1*). Increase *FSHβ* expression establishes a state of photosensitivity to stimulatory daylength, and *TSHβ* thryotropes in the pars tuberalis monitor daylength and when light stimulation occurs during a period of photoinducibility, initiate gonadal development. The two-component model is exciting as it accommodates evidence for endogenous growth of quail gonads in the absence of

photostimulation (*Follett and Sharp, 1969*). Moreover, a TSH-independent programmed change in *FSHβ* expression addresses how seasonal rhythms in tropical, non-photoperiod birds can be regulated (*Gwinner and Dittami, 1990*).

The photoperiodic induction of gonadal growth in quail is dependent on circadian timing mechanisms (*Follett and Sharp, 1969*). However, only a few proximal promoters of photoperiodic genes contain D-box elements required for circadian timing input and include eyes absent-3 (*EYA3*) and *TSHβ*, but not *FSHβ* (*Liddle et al., 2022*). Similarly, E-box elements are only identified in the proximal promoter of *EYA3*. The presence of E- and D-boxes provides a clear molecular mechanism by which the circadian clock can control the long photoperiod-induced expression of these highly photoperiodic genes. Conversely, *FSHβ* expression did not show diurnal variation and instead maintained constitutive expression across long (16L:8D), short (8L:16D), and the 'equinox' (12L:12D) photoperiodic conditions. We used a broad, unbiased bioinformatic approach to identify putative transcriptional bindings sites that may regulate *FSHβ* expression. We identified several potential transcriptional binding proteins and *MEF2* motifs were observed across multiple promoters of genes for transcripts in the pars distalis and pars nervosa of the pituitary gland. However, functional analyses are necessary to establish a causal link between these newly identified signaling pathways (e.g., MEF2) and the seasonal regulation of transcript expression.

The high-dimensionality and high-frequency analyses of seasonal transition in physiology used in this study, facilitated the ability to uncover that photoperiod differentially regulates two endocrine systems: reproduction and energy balance. *POMC* has well-described roles in the neuroendocrine regulation of food intake and body mass (*Yeo and Heisler, 2012*). We found that *POMC* expression increased in response to short photoperiods and was associated with delayed body mass and adipose tissue growth. Interestingly, there was a gradual increase in body mass and adipose tissue mass during the transition for short photoperiod to the spring equinox despite elevated levels of *POMC* expression (i.e., adipose). *POMC* levels did not decrease until after exposure to stimulatory long photoperiods. Given the consistent photoperiod-induced change in *POMC* expression across animals (*Helfer and Stevenson, 2020*), these data provide significant insight into the temporal regulation of the central, and peripheral control of seasonal energy balance. As the closely related European quail (*Coturnix coturnix*) are migratory (*Dorst, 1956*; *Bertin et al., 2007*), the increased fattening observed early in the spring transition may reflect a conserved seasonal physiological response to ensure energy stores are provided for migration.

The integration of environmental cues to time breeding in birds varies between male and female birds (*Ball and Ketterson, 2008*; *Tolla and Stevenson, 2020*). In most temperate breeding males, full reproductive development can be achieved in response to photoperiod cues. The robust change in gonadal growth and involution provides a powerful approach to identify the key neuroendocrine mechanisms that govern the avian photoperiodic response. Despite ovarian changes in response to photoperiodic manipulations, female birds generally require other supplemental cues (e.g., temperature, social cues) to attain full reproductive development (*Wingfield, 1980*). In female, white-crowned sparrows (*Zonotrichia leucophrys*), increased photoperiod induces ovarian development to a pre-breeding state (*Farner et al., 1966*), and supplementary cues, such as temperature, can modify ovarian growth (*Wingfield et al., 1996*; *Wingfield et al., 2003*). In Corsica, two populations of great tits (*Parus major*) differ in egg laying date by up to 1 month despite males from both regions displaying similar timing in reproductive development (*Caro et al., 2006*). As the initial predictive environmental cue (i.e., photoperiod) times reproduction similarly in both male and female birds, the data provided in this paper provides key insights into the fundamental mechanisms that govern transitions in the hypothalamo-pituitary control of reproduction. However, studies that seek to understand how supplementary cues (e.g., temperature) are integrated to fine tune the timing of reproduction will require a focus on female birds.

In conclusion, these studies provide a comprehensive transcriptome dataset that can facilitate ecological studies that seek to uncover the molecular substrates which environmental cues, such as temperature, impact phenological timing and mistiming in birds (*Visser and Gienapp, 2019*). The observation for photoperiod-independent regulation of *FSHβ* expression provides a new cellular mechanism in which supplementary environmental cues, such as temperature, can regulate the timing of seasonal reproduction. For example, the marked population differences in Great tit laying dates in Corsica, despite similar daylength cues, might be driven by local temperatures cues acting on

*FSHβ* expression to advance, or delay follicular maturation. Overall, the data indicate a multi-cellular, multi-neural interval timing mechanism resides in the brain and has significant implications for understanding species-specific seasonal transitions in life histories.

## Materials and methods

### Animals

All Japanese quail were provided by Moonridge Farms, Exeter United Kingdom (moonridgefarms.co.uk). Chicks were raised under constant light and constant heat lamp conditions. Five-week-old male birds were delivered to the Poultry facilities at the University of Glasgow, Cochno Farm in September 2019 and 2020. Both male and female birds respond to changes in photoperiod (*Ball and Ketterson, 2008*; *Farner et al., 1966*). Only males were used in the present studies as the robust change in gonadal volume provides a powerful approach to maintain strong statistical power with fewer animals. Food (50:50 mix of Johnston and Jeff, quail mix & Farm Gate Layers, poultry layers supplemented with grit) and tap water was provided ad libitum. All procedures were in accordance with the National Centre for the Replacement, Refinement and Reduction of Animals in Research ARRIVE guidelines (https://www.nc3rs.org.uk/revision-arrive-guidelines). All procedures were approved by the Animal Welfare and Ethics Review Board at the University of Glasgow and conducted under the Home Office Project Licence PP5701950.

### Study 1 – photoperiod-induced transition in seasonal life-history states

Male quail (*N* = 108) were housed in a summer-like long day (LD) photoperiod (16L:8D). To mimic the autumnal decline and subsequent spring increase in the annual photoperiodic cycle, birds were exposure to a sequential change in daylength from 16L (16a), to 14L (14a), to 12L (12a), to 10L (10a), to 8L, then back to 10L (10v), to 12L (12v), to 14L (14v), and lastly 16L (16v) (*Figure 1*). Each photoperiod treatment lasted for 2 weeks to minimize the impact of photoperiodic history effects (*Stevenson et al., 2012a*). At the end of each photoperiodic treatment a subset of quail (*n* = 12) body mass was used as a measure to pseudo randomly select birds for tissue collection and served to reduce the potential for unintentional bias. Birds were killed by cervical dislocation followed by jugular cut. A jugular blood sample was collected in 50 µl heparinized tubes (Workhardt, UK) and stored at −20°C. Brain and pituitary stalk were rapidly dissected, frozen on powdered dry ice and stored at −80°C. Testes were dissected and weighed to the nearest 0.001 g using the Sartorius microbalance (Sartorius, Germany). The length and width of the testes were measured using callipers and volumes were calculated using the equation for a spheroid ($4/3 × 3.14 × [L/2] × [W/2]2$) (*King et al., 1997*). Body mass was measured using an Ohaus microscale to the nearest 0.1 g. Fat score was assessed using the common scale developed by *Wingfield and Farner, 1978*. The scale range is 0–5, which 0 is no visible fat and 5 indicates bulging fat bodies are present.

### Study 2 – daily rhythms in molecular profiles during solstices and equinoxes

To investigate the daily molecular representation of summer- and winter-like solstices and the autumnal and spring equinoxes, Japanese quail (*N* = 188) were subjected to the same photoperiodic treatments described in Study 1. A subset of quail was pseudo randomly selected after the autumnal 12L:12D (*n* = 47), short day 8L:16D (*n* = 48), spring 12L:12D (*n* = 46), and long day 16L:8D (*n* = 47) treatment conditions. For each of the four photoperiodic treatment conditions, five to six birds were collected shortly after lights on (Zeitgeber time (zt) 0), and then every 3 hr for 24-hr period. This resulted in a high-frequency daily sampling period that included zt0, zt3, zt6, zt9, zt12, zt15, zt18, and zt21. Brain, pituitary gland, and liver were extracted and stored at −80°C. Testes mass was determined as described above.

### Study 3 – endogenous programming of FSHβ in the quail pituitary gland

Male quail (*N* = 24) were housed in LD (16L:8D) photoperiod for 2 days, and then daylength was decreased to 12 hr for 1 day, and then 8L for 6 weeks. A subset of quail was killed by decapitation followed by exsanguination and established the photoregressed 8L group (*n* = 6). A subset of birds

($n$ = 6) was maintained in short day photoperiods for four more weeks (8Lext). This group of birds provided the ability to examine whether an endogenous increase in *FSHβ* expression would occur in constant short day photoperiod condition. The other twelve birds were transitioned to the 10L:14D (10v) light treatment for 2 weeks and another subset of birds were collected ($n$ = 6). The last subset of birds was transitioned to 12L:12D (12v) for 2 weeks and were then killed ($n$ = 6). For all birds, the brain and pituitary glands were dissected and immediately frozen in dry ice and then placed at −80°C. This experimental design provided the ability to examine endogenous changes in pituitary cell function via the 8Lext group, and the photoinduced increase in photosensitivity (i.e., 10v and 12v).

## Hypothalamic and pituitary dissection

During brain extraction the pituitary stalk was severed. The procedure leaves pituitary gland components of the pars intermedia, pars distalis, and pars nervosa resting in the sphenoid bone. To isolate the anterior hypothalamus/preoptic area and the MBH, we used a brain matrix and coordinates based on previously published anatomical locations (*Stevenson et al., 2012b*; *Nakao et al., 2008*, respectively). Brains were placed ventral surface in an upward direction. For the anterior hypothalamus/preoptic area a 2-mm diameter from 2-mm brain slice was collected. The rostral edge of the optic nerve was identified and then a 1-mm cut in the rostral and 1-mm cut in the caudal direction was performed. Brain slices were checked to confirm the presence of the tractus septomesencephalicus in the rostral section and the decussation supraoptica dorsalis in the caudal section. These anatomical regions reliably capture the GnRH neuronal population in birds (*Stevenson and Ball, 2009*). The MBH was isolated from a 2-mm punch from a 3-mm brain slice that spanned the decessation supraoptica dorsalis to the nervus oculomotorius (*Nakao et al., 2008*).

## Ribonucleic acid extraction and quantitative PCR

RNA was extracted from anterior hypothalamus/preoptic area, MBH, and pituitary gland tissue using QIAGEN RNesay Plus Mini Kit (Manchester, UK). Nucleic acid concentration and 260/280/230 values were determined by spectrophotometry (Nanodrop, Thermo Scientific). cDNA was synthesized from 100 ng RNA using SuperScript III (Invitrogen), and samples stored at −20°C until quantitative PCR (qPCR) was performed. qPCRs were conducted using SYBR Green Real-time PCR master mix with 5 μl cDNA and 10 μl SYBR and primer mix. All samples were run in duplicate. PCR primer sequences and annealing temperatures are described in *Supplementary file 2*. qPCRs for mRNA expression in tissue were performed using Stratagene MX3000. qPCR conditions were an initial denature step at 95°C for 10 min. Then, 40 cycles of a denature at 95°C for 30 s, a primer-specific annealing temperature for 30 s, and then an extension phase at 72°C for 30 s. The qPCR reaction was terminated at 95°C for 1 min. A melt curve assay was included to confirm specificity of the reactions. The efficiency and cycle thresholds for each reaction were calculated using PCR Miner (*Zhao and Fernald, 2005*). All samples were assessed based on the Minimum Information for Publication of Quantitative Real-Time PCR guidelines (0.7–1.0; *Bustin et al., 2009*). Actin and glyceraldehyde 3-phosphate dehydrogenase were used as the reference transcripts. The most stable reference transcript was used to calculate fold change in target gene expression.

## Minion transcriptome sequencing

RNA for transcriptome sequencing was extracted using QIAGEN RNeasy Plus Mini Kit (Manchester, UK). RNA concentration and 260/280/230 values were determined by Nanodrop spectrophotometer. Isolated RNA reliably has RNA integrity number values >9.0 for both the MBH and pituitary gland. RNA was synthesized into cDNA using Oxford Nanopore Direct cDNA Native Barcoding (SQK-DCS109 and EXP-NBD104) and followed the manufacturer's protocol. A total of 6 Spot-ON Flow cells (R9 version FLO-MIN106D) were used for each tissue. A single quail was randomly selected from each treatment group so that a single flow cell had $n$ = 9 samples giving a total of $N$ = 56 quail for MBH and pituitary stalk transcriptome sequencing. Transcriptome sequencing was conducted using MinION Mk1B (MN26760, Oxford Nanopore Technologies). Sequencing was performed by MinKNOW version 20.10.3 and Core 4.1.2. The parameters for each sequencing assay were kept to 48 hr, −180 mV voltage, and fast5 files saved in a single folder for downstream bioinformatic analyses (*Source code 1*).

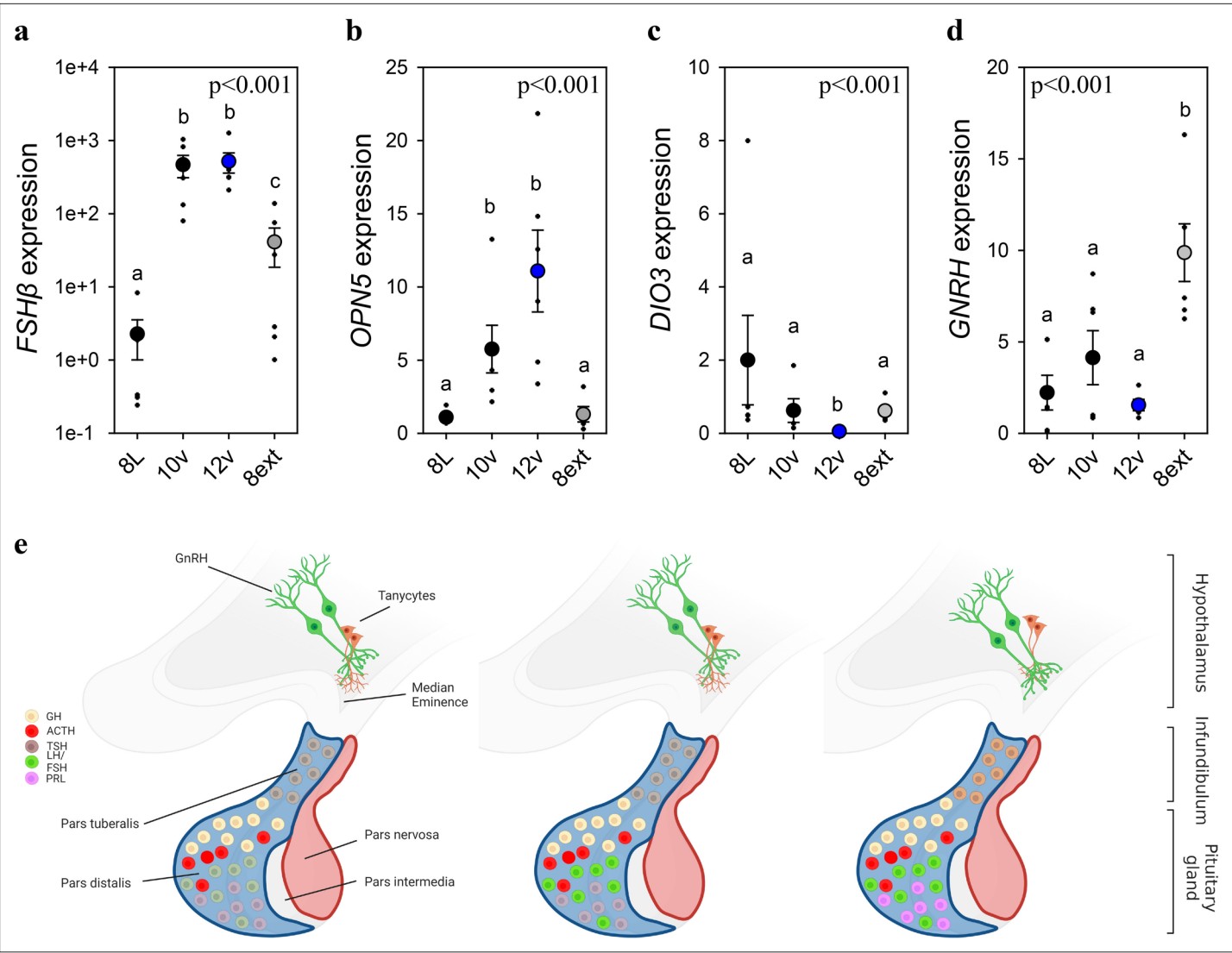

**Figure 3.** Endogenous and light-induced *FSHβ* expression in the pituitary gland. *(a) Follicle-stimulating hormone-β (FSHβ) expression increased during the photoinduced transition from 8L to 10v, and 12v. FSHβ also showed a smaller, yet significantly increased in expression after prolonged exposure to 8L. Y-axis is presented in log-scale due to the significant increase in FSHβ expression in 10v and 12v. (b) OPN5 was detected in the pituitary gland and showed a significant increase in expression in the transition to 10v and 12v, similar to FSHβ expression. (c) DIO3 was significantly reduced in 12v quail compared to all other treatment groups. (d) GNRH expression remained constant during the transition from 8L to 12v. However, continued exposure to 8L was observed to increase GNRH expression.* Data are mean ± standard error of the mean (SEM) and residual dot plot (**a–d**). One-way analysis of variance (ANOVA) with Tukey's test for multiple comparisons. Letters denote significant difference between photoperiod phases. Raw data are available in *Figure 1—source data 1*. (**e**) Schematic representation of the endogenous and light-dependent increase in pituitary cell types during the transition from 8L to stimulatory 16v light treatments. Increased color indicates increased transcript expression.

The online version of this article includes the following figure supplement(s) for figure 3:

**Figure supplement 1.** Endogenous and light-induced molecular switches in the quail mediobasal hypothalamus and pituitary gland.

## Transcriptome analyses

The transcriptome data analysis pipeline is outlined in ***Supplementary file 3***. All bioinformatic steps were conducted using R Studio and run in a Conda environment. First, fast5 files were demultiplex and basecalled by Guppy 4.2.1. Then Porechop v0.2.4 was conducted to remove adapters from reads followed by Filtlong v0.2.0 to filter long reads with minimum 25 bases and mean *q* weight of 9. Transcripts were aligned to the Japanese quail reference genome and transcriptome using Minimap2 v2.17 (*Li and Birol, 2018*). Transcript expression levels were determined using Salmon v0.14.2 and EdgeR v3.24.3 for normalization and differential expression (*Patro et al., 2017*). DAVID was conducted to

identify functional pathways active during the transitions across photoperiodic states (*Figure 1—figure supplement 1*; *Figure 1—source data 4*; *Dennis et al., 2003*).

## BioDare2.0 analyses of significant differentially expressed genes

To identify seasonal rhythmic expression of transcripts, we selected differentially expressed genes identified by EdgeR (p < 0.05). Data were analyzed using nonlinear regression for rhythmicity using the online resource BioDare 2.0 (*Zielinski et al., 2014*) (biodare2.ed.ac.uk). The empirical JTK_CYCLE method was used for detection of rhythmicity and the classic BD2 waveform set was used for comparison testing. The type of transcript rhythmicity was confirmed as (e.g., sine/cos/arcsine) or non-rhythmic (spike) expression. Rhythmicity was determined by a Benjamini–Hochberg controlled p-value (BH corrected p < 0.1). Data for heatmaps were clustered using PAM clustering from the cluster package (). Heatmaps were created using the Complexheatmaps package (*Gu et al., 2016*). Heatmaps generated from statistically significant transcripts are presented in *Figure 1—source data 3 and 5*.

## Enrichment factor identification in promoters of differentially expressed genes

To explore potential upstream molecular pathways involved in the regulation of differentially expressed genes, we conducted a transcription factor analyses. The aim was to identify which transcription factor or factors are responsible for observed changes in gene expression and whether, if any overlap occurs across tissues. Transcription factor enrichment analysis was achieved using ChIP-X Enrichment Analysis 3 (ChEA3) (*Keenan et al., 2019*). We used a conservative approach and only used transcripts detected as significant by BioDare 2.0 analysis (BH corrected p < 0.05). Enriched transcription factors were ranked using the ENCODE database and presented in *Figure 1—source data 7 and 8*.

## Weighted gene co-expression network analyses

Co-expression networks were established using WGCNA package in R (*Langfelder and Horvath, 2008*). Raw data from pituitary and MBH sequencing were filtered to remove lowly expressed transcripts identified using EdgeR. Data were assessed for outliers and values were excluded. The data also were assessed for scale independence and mean connectivity, and a power threshold of 5 was selected. The WGCNA package was used to construct a weighted gene network, with a merging threshold of 0.25. Module–trait relationship associations were used to identify relationships with measured physiological data. Data for the analyses are provided in *Figure 1—source data 6*.

## Daily waveform analyses of transcript expression

To establish rhythmicity in daily waveform expression of MBH and pituitary transcripts, we conduct cosinor analyses (*Cornelissen, 2014*). 2-ΔΔ Cycling time (Ct) values obtained for the genes investigated in Study 2 were subjected to cosinor analyses based on unimodal cosinor regression [$y = A + (B \dot{c} \cos(2\pi(x - C)/24))$], where $A$, $B$, and $C$ denote the mean level (mesor), amplitude, and acrophase of the rhythm, respectively. The significance of regression analysis determined at $p < 0.05$ was calculated using the number of samples, $R^2$ values, and numbers of predictors (mesor, amplitude, and acrophase) (*Singh et al., 2013*). Data are compiled in *Figure 2—source data 1*.

## Bioinformatic analyses of FSHβ promoter

To identify potential links between transcription factors identified using bioinformatic tests, and transcriptome data, we examined binding motifs in the FSHβ promoter. The upstream promotor sequence of 3500 bp (−3500 to 0) of Japanese quail FSHβ was obtained from Ensemble (http://www.ensembl.org/index.html). This promotor sequence was analyzed by CiiDER transcription-binding factor analysis tool (*Gearing et al., 2019*) against JASPER core database. DNA motifs in the promoter were unique for 470 transcription-binding factors. The top 80 transcription-binding factors were then subjected to PANTHER gene ontology enrichment analyses (*Mi et al., 2013*). Several pathways were discovered and included hormone responsive, epigenetic, and responsive to nutrients (*Figure 2—source data 2*).

## Immunocytochemistry and histological analyses

Snap-frozen brains from experiment 1 were sectioned at 20 μm using a cryostat (CM1850, Leica). The tissues sections were collected in supercharged slides (631-0108, VWR) and stored in −80°C till

processed for staining. We used mouse monoclonal antibody (OAAEE00561, Aviva systems) raised against the Human vimentin gene (NCBI Reference Sequence: NP_003371.2). The antibody has been shown to specifically bind with vimentin expressed in avian cells (https://www.avivasysbio.com/vim-antibody-oaee00561.html). Human and quail vimentin peptide show high similarity of 86.96% similarity. In silico analyses using BLAST confirmed that the antibody sequence is specific to vimentin. The next closest protein had 68% similarity (i.e., desmin) which is expressed in cardiac and skeletal muscle. We used Goat Anti-mouse Alexa Flour488 (A11001, Invitrogen) for the secondary antibody. As negative control procedures, we performed omission of primary antibody and omission of secondary antibody.

Immunocytochemistry was performed using the standard immunofluorescence protocol (*Majumdar et al., 2015*) with minor modifications. Briefly, sections (brain) in slides were first enclosed in margin using ImmEdge pen (H-4000, Vector Labs). The sections were then first post fixed in 10% neutral buffered formalin (5735, Thermo Scientific) for 4 hr. After fixing, the sections were washed (three times; 5 min each) with TBS (phosphate buffer saline with 0.2% Triton). Then they were blocked in 20% bovine serum albumin in TBS for 1 hr at room temperature (RT). Subsequently the blocking solution was removed by pipetting and the sections were incubated with primary antibody (1:300 dilution) for 2 hr at RT and finally overnight at 4°C. The next day, sections were first washed with TBS (three times; 5 min each) and then incubated with secondary antibody (1:200 dilution) for 2 hr at RT. Finally, the sections were again washed with TBS (three times; 5 min each) and mounted in Fluromount-G mounting media with DAPI (4',6-diamidino-2-phenylindole; 004959-52, Invitrogen). The dried slides were visualized using Leica DM4000B fluorescence microscope equipped with Leica DFC310 FX camera. Leica Application Suite (LAS) software was used for image acquisition. All the images were taken at constant exposure for the FITC channel at ×10 and ×20 magnification.

For analysis and quantification of % area, ImageJ version 1.53j was used. For this, ×20 images were first converted into greyscale images (8 bit) and a threshold applied. The threshold was determined using the triangle method on multiple randomly selected images and applied for all the images. The scale of measurement in ImageJ was then set to CM in 300 pixels/cm scale. A region of interest in the median eminence and dorsal 3rdV ependymal layer was specified as $300 \times 500$ pixels ($1 \times 1.67$ cm scaled units) and area fraction was measured. At least three images from each animal were measured and averaged for each bird. A total of 27 quail were used and distributed evenly across photoperiod treatments ($n = 3$).

## Statistical analyses and figure presentation

GLM tests were conducted to test for statistical significance. One-way analysis of variance (ANOVA) with Bonferroni correction was applied to testes mass, body mass, fat score, vimentin immunoreactivity, and qPCR analyses in Study 1. Two-way ANOVAs with photoperiod and zeitgebers main effects were conducted on qPCR analyses in Study 2. One-way ANOVAs were conducted for qPCR data in Study 3. qPCR data were log-transformed if violation of normality was detected. Significance was determined at p < 0.05. Figures were generated using AdobeIllustrator and BioRender was used to create images in panels (*Figures 1d, n and 3e* and *Figure 3—figure supplement 1*).

## Acknowledgements

The authors thank Elisabetta Tolla, Christopher Elcombe, Ana Monteiro, and David Hamilton for their assistance. The authors thank Professor Gregory Ball and Neil Evans for comments on a previous version of the paper. Funding: the work was funded by a Leverhulme Trust Research Leader to TJS.

## Additional information

### Funding

| Funder | Grant reference number | Author |
|---|---|---|
| Leverhulme Trust | RL-2019-06 | Tyler Stevenson |

| Funder | Grant reference number | Author |
| --- | --- | --- |

The funders had no role in study design, data collection, and interpretation, or the decision to submit the work for publication.

## Author contributions

Gaurav Majumdar, Data curation, Formal analysis, Validation, Investigation, Visualization, Methodology, Writing - original draft; Timothy A Liddle, Validation, Investigation, Visualization, Methodology; Calum Stewart, Data curation, Formal analysis, Validation, Investigation, Visualization, Methodology; Christopher J Marshall, Investigation, Methodology; Maureen Bain, Supervision, Investigation, Methodology, Writing - review and editing; Tyler Stevenson, Conceptualization, Data curation, Formal analysis, Supervision, Funding acquisition, Visualization, Writing - original draft, Project administration, Writing - review and editing

## Author ORCIDs

Christopher J Marshall  http://orcid.org/0000-0002-5658-9817
Tyler Stevenson  http://orcid.org/0000-0003-2644-9685

## Ethics

All procedures were in accordance with the National Centre for the Replacement, Refinement and Reduction of Animals in Research ARRIVE guidelines (https://www.nc3rs.org.uk/revision-arrive-guidelines). All procedures were approved by the Animal Welfare and Ethics Review Board at the University of Glasgow and conducted under the Home Office Project Licence PP5701950.

Reviewer #1 (Public Review): https://doi.org/10.7554/eLife.87751.3.sa1
Reviewer #2 (Public Review): https://doi.org/10.7554/eLife.87751.3.sa2
Author Response https://doi.org/10.7554/eLife.87751.3.sa3

# Additional files

## Supplementary files

• Supplementary file 1. Two distinct pituitary cell types are involved in the external coincidence model for the avian photoperiodic response. Short photoperiods activate an endogenously generated programmed to increase follicle-stimulating hormone-β ($FSH\beta$) expression in the pars distalis of the pituitary gland. A gradual increase in non-stimulatory photoperiods, such as 10v and 12v, establish the photosensitive state and characterized by constituently expressed $FSH\beta$. Photoperiods that extend beyond the critical daylength (e.g., >12v) activates thyrotropes thyrotropin-stimulating hormone-β ($TSH\beta$) expression in the pars tuberalis of the pituitary gland. The coincidence timing of long day $TSH\beta$ with the short day photosensitivity induced by increased $FSH\beta$ expression results in gonadal development. Figure was created using BioRender.

• Supplementary file 2. Quantitative PCR primer and parameter.

• Supplementary file 3. Bioinformatic pipeline.

• MDAR checklist

• Source code 1. R-code for analyses.

## Data availability

All raw data are available in *Figure 1—source data 1*. Raw sequencing data are available in Gene expression omnibus database GSE241775 and BioProject PRJNA1009845. R code used is available in Source code 1.

The following datasets were generated:

| Author(s) | Year | Dataset title | Dataset URL | Database and Identifier |
|---|---|---|---|---|
| Stevenson T | 2023 | FSHβ links photoperiodic signalling in the mediobasal hypothalamus and pituitary gland to seasonal reproduction in Japanese quail | https://www.ncbi.nlm.nih.gov/geo/query/acc.cgi?acc=GSE241775 | NCBI Gene Expression Omnibus, GSE241775 |
| University of Glasgow | 2023 | FSHβ links photoperiodic signalling in the mediobasal hypothalamus and pituitary gland to seasonal reproduction in Japanese quail | https://www.ncbi.nlm.nih.gov/bioproject/PRJNA1009845/ | NCBI BioProject, PRJNA1009845/ |

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
