## [Editor Report · eLife assessment]

This **important** article provides insights into the neural centers and hormonal modulations underlying seasonal changes associated with photoperiod-induced life-history states in birds. The physiological and transcriptomic analyses of the mediobasal hypothalamus and pituitary gland offer evidence for a compelling timing mechanism for measuring day length, which is relevant for the field of seasonal biology. The study's **convincing** experiments and findings have the potential to captivate the attention of molecular and organismal endocrinologists and chronobiologists.

---

## [Referee Report · Reviewer #1 (Public Review)]

This study is carefully designed and well executed, including a comprehensive suite of endpoint measures and large sample sizes that give confidence in the results. The authors have satisfactorily addressed my concerns. Specifically, the new graphical description of the experimental design along a timeline will be very helpful in guiding the reader through the paper. The narrative style is much improved and highly technical terminology is minimized. The authors now also address the question of sex differences, which will be important to study in future research. The additional analyses carried out by the authors are illuminating.

---

## [Referee Report · Reviewer #2 (Public Review)]

It is well known that as seasonal day length increases, molecular cascades in the brain are triggered to ready an individual for reproduction. Some of these changes, however, can begin to occur before the day length threshold is reached, suggesting that short days similarly have the capacity to alter aspects of phenotype. This study seeks to understand the mechanisms by which short days can accomplish this task, which is an interesting and important question in the field of organismal biology and endocrinology.

The set of studies that this manuscript presents is comprehensive and well-controlled. Many of the effects are also strong and thus offer tantalizing hints about the endo-molecular basis by which short days might stimulate major changes in body condition. Another strength is that the authors put together a compelling model for how different facets of an animal's reproductive state come "on line" as day length increases and spring approaches. In this way, I think the authors broadly fulfill their aims.

---

## [Author Response]

The following is the authors’ response to the original reviews.

**Editorial comments:**
Comment 1 - Recommendations for the authors: please note that you control which revisions to undertake from the public reviews and recommendations for the authors.

We appreciate the feedback from the 3 Reviewers and Editor. We have enumerated each Reviewer comment and provide a detailed response. We endeavoured to include each suggestion into the revised manuscript. All changes in the manuscript are indicated in red font. In instances in which we respectfully disagree with the Reviewer, we have provided a fair rebuttal. We feel the comments from the Reviewers has significantly improved the clarity and quality of the manuscript.

Comment 2 - The revision process has demonstrated the value of your work, highlighting both its strengths and shortcomings. Importantly, it provides detailed and achievable suggestions for improving the current version of your contribution.

We thank the Reviewers and Editor for their time and expert input on our manuscript. We feel the suggestions from the Reviewers to address the shortcomings has resulted in a significantly improved manuscript.

Comment 3 - There is a general consensus among the reviewers on three key aspects. Firstly, the article would greatly benefit from a clearer layout of the experimental design and methodology, potentially including schematics to help readers comprehend the complexity and details of the study.

We appreciate the feedback from Reviewer 2 in particular. We have added a new schematic for Experiment 3 (see PUBLIC REVIEWS Reviewer #2 Comment 2). We have also revised the Results section by including subheadings and additional text to help explain the methods.

Comment 4 - Secondly, conducting a more comprehensive analysis of the available dataset, utilizing tools such as WGCNA to explore gene co-expression networks beyond specific genes, is recommended. Additionally, it is advised to exercise greater caution when discussing the limitations of the employed methods.

The suggestion for the WGCNA is excellent and very much appreciated. The revised manuscript includes WGCNA for both the MBH and pituitary gland. (See Figures S3 & Table S6 and lines 166-182; 497-505).

Comment 5 - Thirdly, expanding the results section to create a more engaging narrative that guides readers through the numerous findings, and extending the discussion and conclusions to emphasize the ecological relevance of learning photoperiodic/seasonal responses and highlighting the presented model, would be valuable.

These were excellent suggestions that significantly improved the clarity and quality of the manuscript. The results section included several subheadings to help break up of the transitions across experiments. We have also significantly revised the introduction and discussion to include the ecological relevance and importance to consider sex as a factor in the interpretations.

Comment 6 - Finally, please pay close attention to the comment on the statistical analysis provided by Rev#2.

It is unclear why the Benjamini-Hochberg’s FDR analyses was suggested. The statistical test is a version of the Bonferroni test but is less stringent. We prefer to use conservative tests (i.e., Bonferroni correction). Moreover, the Bonferroni correction is the commonly used statistical tests in the field. To be consistent with the field and to be careful in our statistical approach, the revised manuscript did not change the post-hoc correction.

**PUBLIC REVIEWS:**

**Reviewer #1:**
Comment 1 - The authors investigated the molecular correlates in potential neural centers in the Japanese quail brain associated with photoperiod-induced life-history states. The authors simulated photoperiod to attain winter and summer-like physiology and samples of neural tissues at spring, and autumn life-history states, daily rhythms in transcripts in solstices and equinox, and lastly studies FSHb transcripts in the pituitary. The experiments are based on a series of changes in photoperiod and gave some interesting results. The experiment did not have a control for no change in photoperiod so it seems possible that endogenous rhythms could be another aspect of seasonal rhythms that lack in this study. The short-day group does not explain the endogenous seasonal response.

We thank the Reviewer for the fair assessment of the manuscript. The statement ‘the experiment did not have a control for no change in photoperiod’ is not clear to us. We think the Reviewer is arguing that prolonged constant photoperiod was not conducted to examine circannual timing in avian reproduction. The constant short photoperiod in Exp3 does provide the ability to examine the initial stages of interval timing. A different endogenous mechanism used by animals. The revised manuscript has clarified the different physiological responses.

Comment 2 - The manuscript would benefit from further clarity in synthesizing different sections. Additionally, there are some instances of unclear language and numerous typos throughout the manuscript. A thorough revision is recommended, including addressing sentence structure for improved clarity, reframing sentences where necessary, correcting typos, conducting a grammar check, and enhancing overall writing clarity.

We have incorporated the suggestions from both Reviewer 1 and Reviewer 2 that aimed to increase the clarity of the manuscript. We have provided detailed responses to each comment below and state how each comment was incorporated in the revised manuscript. We also had the manuscript reviewed by a colleague to help identify issues associated with sentence structure, grammar, and spelling.

Comment 3 - Data analysis needs more clarity particularly how transcriptome data explains different physiological measures across seasonal life-history states. It seems the discussion is built around a few genes that have been studied in other published literature on quail seasonal response. Extending results on the promotor of DEGs and building discussion is an extrapolating discussion on limited evidence and seems redundant.

A new statistical analysis (ie., WGCNA) was conducted to identify relations between photoperiod, physiology and transcripts. The focus on the few photoperiodic gene was kept in the discussion as the transcript expression is important to highlight the differences from the prevailing hypotheses and novel patterns of expression across seasonal timescales. (See Figures S3 & Table S6 and lines 166-182; 497-505).

Comment 4 - Last, I wondered if it would be possible to add an ecological context for the frequent change in the photoperiod schedule and not take account of the endogenous annual response. Adding discussion on ecological relevance would make more sense.

This is an excellent suggestion. The introduction and discussion were substantially revised to include the ecological relevance.

**Reviewer #2:**
Comment 1 - This study is carefully designed and well executed, including a comprehensive suite of endpoint measures and large sample sizes that give confidence in the results. I have a few general comments and suggestions that the authors might find helpful.

We appreciate the Reviewers support for our manuscript. We have endeavoured to incorporate all suggestions in the revised manuscript.

Comment 2 - I found it difficult to fully grasp the experimental design, including the length of light treatment in the three different experiments (which appears to extend from 2 weeks up to 8 weeks). A graphical description of the experimental design along a timeline would be very helpful to the reader. I suggest adding the respective sample sizes to such a graphic, because this information is currently also difficult to keep track of.

We have created a new figure panel to address the Reviewer’s concern. See figure S4 panel ‘a’. The new schematic representation was designed to illustrate the similarity in experimental design used in Experiment 1 and Experiment 2. But clearly illustrates the extended short photoperiod manipulation (4 weeks and not 8 weeks). We added the sample sizes to initial drafts but felt the added text hindered the clarity of the schematic representation (particularly for Fig1a). The sample sizes for each experiment and treatment are provided in the raw data provided in the supplementary Table 1. For this reason, we have opted to not add the sample size to each diagram. We hope that the Reviewer will understand our perspective.

Comment 3 - The authors use a lot of terminology that is second nature to a chronobiologist but may be difficult for the general reader to keep track of. For example, what is the difference between "photoinducibility" and "photosensitivity"? Similarly, "vernal" and "autumnal" should be briefly explained at the outset, or maybe simply say "spring equinox" and "fall equinox."

This is a very helpful suggestion, and we thank the Reviewer. Two changes were made to the manuscript to address this comment. First, we revised the second introductory paragraph to describe the photoperiodic response and the terms used. Second, we have removed all reference to ‘vernal’ and replaced with ‘spring’. We opted to keep ‘autumn’ as the change to ‘fall’ did not provide the clarity of seasonal state in some statements (as fall is also used as a downward direction).

Comment 4 What was the rationale for using only male birds in this study? The authors may want to include a brief discussion on whether the expected results for females might be similar to or different from what they found in males, and why.

We agree with the Reviewer’s position that studies should include, or least describe, male and female biology. We have revised the text to address this comment. In the methods, we provide 2 sentences that state the photoperiodic response is the same for both male and females, and why males were selected. See lines (352-355). Then, in the discussion, we describe why females will be important to study how other supplementary environmental cues impact seasonal timing of reproduction. See lines (312-330; and 334-339).

Comment 5 - The authors used the Bonferroni correction method to account for multiple hypothesis testing of measures of testes mass, body mass, fat score, vimentin immunoreactivity and qPCR analyses in Study 1. I don't think Bonferroni is ever appropriate for biological data: these methods assume that all variables are independent of each other, an assumption that is almost never warranted in biology. In fact, the data show clear relationships between these endpoint measures. Alternatively, one might use Benjamini-Hochberg's FDR correction or various methods for calculating the corrected alpha level.

This concern is not clear to us. The Benjamini-Hochberg’s FDR is a slight modification of the Bonferroni correction. Moreover, the FDR is a less-stringent statistical test compared to the Bonferroni correction. We prefer to keep the Bonferroni approach to correct for multiple tests for two reasons. First, this test is commonly used in the field of chronobiology, and second, the Bonferroni correction is more conservative. We hope the Reviewer will appreciate our perspective to be consistent with the research field and higher stringency in our statistical approach.

Comment 6 - The graphical interpretations of the results shown in Figure 1n and Figure 3e, along with the hypothesized working model shown in Figure S5, might best be combined into a single figure that becomes part of the Discussion. As is, I do not think these interpretative graphics (which are well done and super helpful!) are appropriate for the Results section.

We appreciate the Reviewer’s suggestion. During the revision we developed a single figure to show the graphical representation for the respective experiments. Unfortunately, we found the single source to be very difficult to provide a clear description and overview of the findings. We feel that the interpretations, (admittedly unusual for Results section) are best placed in the respective figures that correspond to the different experiments.

**Reviewer #3:**
Comment 1a - It is well known that as seasonal day length increases, molecular cascades in the brain are triggered to ready an individual for reproduction. Some of these changes, however, can begin to occur before the day length threshold is reached, suggesting that short days similarly have the capacity to alter aspects of phenotype. This study seeks to understand the mechanisms by which short days can accomplish this task, which is an interesting and important question in the field of organismal biology and endocrinology.

We thank the Reviewer for their positive feedback.

Comment 1b - The set of studies that this manuscript presents is comprehensive and well-controlled. Many of the effects are also strong and thus offer tantalizing hints about the endo-molecular basis by which short days might stimulate major changes in body condition. Another strength is that the authors put together a compelling model for how different facets of an animal's reproductive state come "on line" as day length increases and spring approaches. In this way, I think the authors broadly fulfill their aims.

We thank the Reviewer for the positive support of our research and manuscript.

Comment 1c - I do, however, also think that there are a few weaknesses that the authors should consider, or that readers should consider when evaluating this manuscript. First, some of the molecular genetic analyses should be interpreted with greater caution. By bioinformatically showing that certain DNA motifs exist within a gene promoter (e.g., FSHbeta), one is not generating robust evidence that corresponding transcription factors actually regulate the expression of the gene in question. In fact, some may argue that this line of evidence only offers weak support for such a conclusion. I appreciate that actually running the laboratory experiments necessary to generate strong support for these types of conclusions is not trivial, and doing so may even be impossible. I would therefore suggest a clear admission of these limitations in the paper.

We agree with the Reviewer’s position. The transcription binding protein analyses was used as a means to identify potential factors involved in the regulation of transcript expression. We have written a new paragraph to address this comment. In the discussion, we that highlight the links between the well characterised circadian regulation of photoperiodic transcripts (e.g, D- & E-box elements and the photoperiodic control of TSHβ). We also indicate that our bioinformatic approach identified potentially new transcription binding motifs, and provide a clear admission and state that functional analyses are required to determine necessity of these pathways (e.g., MEF2). See lines 293-295.

Comment 2 - Second, I have another issue with the interpretation of data presented in Figure 3. The data show that FSHbeta increases in expression in the 8Lext group, suggesting that endogenous drivers likely act to increase the expression of this gene despite no change in day length. However, more robust effects are reported for FSHbeta expression in the 10v and 12v groups, even compared to the 8Lext group. Doesn't this suggest that both endogenous mechanisms and changes in day length work together to ramp up FSHbeta? The rest of the paper seemed to emphasize endogenous mechanisms and gloss over the fact that such mechanisms likely work additively with other factors. I felt like there was more nuance to these findings than the authors were getting into.

We agree with the Reviewer and a similar concern was raised by Reviewer 1. Our aim was to highlight that FSH expression increased in constant short photoperiod. We have revised the manuscript to address the concern raised by the Reviewer. We have added 2 sentences in the results to highlight the additive role of endogenous timing and photoperiodic effects on FSH expression (see lines 223-226). We have kept the text that describes endogenous increases in expression (e.g., FSH/GnRH) in response to short photoperiod in the manuscript as this observation is not influenced by long photoperiod.

Comment 3 - Third, studies 1 - 3 are well controlled; however, I'm left wondering how much of an effect the transitions in day length might have on the underlying molecular processes that mediate changes in body condition. While the changes in day length are themselves ecologically relevant, the transitions between day length states are not. How do we know, for example, that more gradual changes in day length that occur over long timespans do not produce different effects at the levels of the brain and body? This seemed especially relevant for study 3, where animals experience a rather sudden change in day length. I recognize that these experimental methods are well described in the literature, and they have been used by endocrinologists for a long time; nonetheless, I think questions remain.

There are two points raised in this comment. First, the effect of transition in day length on body condition. We are investigating the impact of photoperiodic transitions on body condition. The ongoing project has examined the changes in tissue lipid content and conducted transcriptomic analyses of multiple peripheral tissues involved in energy balance. Although we made an initial attempt to combine all the findings into a single manuscript, the large datasets resulted in an overwhelming manuscript that lacked clarity. Instead, we have opted for two manuscripts that focus on the respective physiological systems. Those data should be published shortly. We did expand the discussion by developing a single paragraph that focused on the pattern of POMC expression and changes in quail body mass and adipose tissue. See lines 300-311.

Second, the Reviewer raised the issue of more gradual changes in day length over longer timespans. The day length and duration of exposure selected was to replicate previously used photoperiod manipulations to ensure reproducibility in research programmes, and to reduce the impact of photoperiod history (see lines 367-369). The present manuscript is the first study in birds to examine multiple intervening (ie within the extreme long- and short-photoperiods) day length conditions and we feel this is a major and novel contribution to the field. We agree that other time points (e.g., 13L:11D), or quicker/longer timespans could provide additional insight into the molecular mechanisms that govern seasonal transitions in reproduction/energy balance. The question raised by the Reviewer requires the types of studies that use natural conditions from wild-caught animals (or semi-natural laboratory settings) and beyond the focus of the current manuscript.

**Recommendations For The Authors:**

**Reviewer #1**
Comment 1 - Abstract: Overall abstract needs more clarity in rationale, hypothesis, and result outcomes. How this study advances our knowledge in seasonal/ photoperiodic regulation of reproduction in birds. Particularly what knowledge gap FSHb results fill in.

We have substantially revised the abstract considering the Reviewer’s suggestions. The abstract has clarified the rationale, hypothesis and results outcomes. We have also added new introductory and concluding statements that place the work into a wider ecological context (as suggested below).

Comment 2 - In general the introduction needs more clarity and doesn't seem to cover the ecological relevance of learning photoperiodic/seasonal response.

We agree with the Reviewer the introduction could be improved. We have substantially revised the introduction with an aim to increase the clarity. This involved an addition on the ecological context, clarification of the photoperiodic states in birds, and a description of the general and specific objectives. Note we did not include an introduction to ‘learning’ of the photoperiodic response, as the term implies a cognitive component is involved which is incorrect. See lines (61-67, 71-74, 80-86, and 100-105).

Comment 3 - Line 58: What does the author mean by "future seasonal environment" Is it to introduce change in climate or future seasonal events? This sentence needs rephrasing and more clarity.

In response to Comment 2, we have revised the introductory paragraph and the sentence was removed from the text.

Comment 4 - Line 63: I would recommend the use of circannual rhythms with caution for the kind of experiments authors have proposed. The approach used here is beyond the scope of addressing circannual endogenous rhythms, which can be tested only independent of photoperiod change.

We agree with the Reviewer’s concern. The use of circannual rhythms was limited to the first paragraph (lines 56-63) only to introduce the concept of endogenous rhythmicity. We were careful to not use the term ‘circannual’ for the rest of the manuscript, as the Reviewer has indicated, would be inappropriate. We have retained the use of ‘endogenous program’ to refer to the molecular and physiological changes that can occur independent of photoperiod change (ie Experiment 3). In this case, the use of endogenous is appropriate as this form of timing adheres to an interval timer. We also provided a definition for interval timer and ecological examples to illustrate the difference between circannual rhythms and annual interval timer (see lines 71-74). We also reviewed the entire manuscript to ensure the distinction for the endogenous program was clear.

Comment 5 - Another aspect authors missed is that Quail is not an absolute photorefractory (Robinson and Follett, 1982).

We agree with the Reviewer that quail are not absolute photorefractory (but instead relative photorefractory). As our photoperiod manipulations do not address criterion 1, or criterion 2 of the avian photoperiodic response (MacDougall-Shackelton et al., 2009; see https://doi.org/10.1093/icb/icp048), we feel that adding the type of photorefractory response would be a distraction and reduce the clarity of the concepts/experimental design described in the manuscript.

Comment 6 - Line 223-234: "Chicks were raised under constant light and constant heat lamp". Constant photoperiod experienced during development raises concern on how this pretreatment would shape the adult seasonal response, which could be different in the seasonal response of birds raised in natural photoperiod. If this is correct, the results shown are not tenable for birds inhabiting the natural environment.

The light schedule used in our experiment is the most appropriate for laboratory reared chicks. The light schedule, use of an incubator and hatchery is commonly used in research laboratories. The procedure serves to increase the hatch rate and welfare of chicks. Undoubtedly there will be some early developmental programming effects on quail development. However, the gonadal response across all 3 experiments was consistent with the vast scientific literature on the avian photoperiodic response in both laboratory and wild birds. As the robust gonadal response clearly replicated previous studies, we are confident the results are tenable for birds inhabiting natural environments.

Comment 7 - Numerous studies done in mammals suggest that photoperiod experienced in the early life stage affects the circadian and seasonal response in adults (Ciarleglio et al., 2011, Perinatal photoperiod imprints the circadian clock, Nat Neurosceince; Stetson M., et al., 1986, Maternal transfer of photoperiodic information influences the photoperiodic response of prepubertal Djungarian hamsters).

We agree with the Reviewer that developmental programming in mammals is important for the photoperiodic response. However, there are vast differences between the avian and mammalian photoperiodic response. Critically, in mammals, the maternal transfer of information to the offspring is achieved via the melatonin hormone. Conversely, in birds, melatonin is not necessary, nor sufficient for photoperiodic time measurement (Juss et al., 1993; see https://doi.org/10.1098/rspb.1993.0121). It is not scientifically tenable to relate the mammalian and avian photoperiodic responses in adulthood based on early developmental programs. For this reason, we did not introduce or discuss developmental programming in our manuscript.

Comment 8 - Please give details on the month in which these birds were exposed to different short and long photoperiods. It is not clear in the method section. The birds experience long to short day transition and then back to long day in 16 weeks (~ 4 months). The annual cycle is ~12 months long in nature. Again, what is the ecological relevance of such an experimental paradigm. This could give some idea on photoperiodic response, but not on how the endogenous annual cycle would respond.

Birds were delivered in September 2019 and 2020. We have added these details to the manuscript (see lines 351-352). We agree with the Reviewer that the ecological relevance of the experimental design is limited. Our focus was to use laboratory conditions and well characterised photoperiodic manipulations to examine the role of the environmental, initial predictive cue to time seasonal transitions in reproduction. The 2-week duration for each photoperiod state in Experiment 1 provides the ability to eliminate the impact of photoperiodic history (see lines 367-369; Stevenson et al., 2012a) and reduce the time necessary for the research project. As described above in Comment #4 – we did not examine the endogenous annual cycle – but instead focused on an endogenous interval timer. Experiment 3 was designed to best examine an endogenous interval timer.

Comment 9 - Line 251: "A jugular blood sample" Please rephrase this sentence and add 50 ul heparinized tubes

We thank the Reviewer for identifying this oversight. The text was changed accordingly.

Comment 10 - Line 259: The scale.....fat pads" - The sentence doesn't read correctly.

The sentence was revised accordingly.

Comment 11 - Line 274: Male.....six weeks. It is not clear from this sentence; what photoperiod birds were exposed to before transferring to 2 long days. Is it 16 or 14 LD.

The birds were held in 16L. The text has been revised accordingly.

Comment 12 - Line 276: It is not clear what is Home Office approved schedule 1. This may be a commonly used term for animal sacrifice protocol in UK and Europe. But it is not familiar jargon for the rest of the globe.

We apologise for the jargon. The text was revised to include the exact methods (decapitation followed by exsanguination).

Comment 13 - Line 277-284: Birds under SD for 4 weeks (8 Lext) is a bit confusing and particularly in the context of studying endogenous rhythm. Needs more clarity.

The text was revised to improve the clarity. The manuscript now states: ‘A subset of birds (n=6) was maintained in short day photoperiods for four more weeks (8Lext). This group of birds provided the ability to examine whether an endogenous increase in FSHβ expression would occur in constant short day photoperiod condition.’

Comment 14 - Line 322-323: Give RIN number (RNA integrity number) here which is a very common parameter to determine RNA degradation in RNAseq experiments. I guess, the MiniON is a portable sequencer and sequences one sample at a time. If this is true authors should consider any batch effect in sequencing and use it as a covariate in the model.

The RIN values from our extraction protocol reliably produce RIN values >9.0. The text now states: Isolated RNA reliably has RIN values >9.0 for both the mediobasal hypothalamus and pituitary gland. Our RIN values are well above the recommended 7.0 limit. The Reviewer is correct that MinION is portable, however, more than one sample can be run at a time. We stated in the text (lines 454-460) that birds were counterbalanced across Flow cells so that each sequencing run had 9 samples, one from each treatment group. Our counterbalancing approach and quality control steps prevented batch effects.

Comment 15 - Line 397-398: Adding quail or chicken-specific vimentin peptide pre-incubation with primary Ab will serve more confirming control. Omitting primary Ab doesn't address cross-reactive/ nonspecific binding issues.

We agree that a positive control (ie primary Ab) is the gold standard to support specificity of the antibody. Unfortunately, we have not found a supplier of the epitope for quail/chicken vimentin. We have conducted another in silico analysis an established that the sequences for the vimentin antibody is specific for vimentin. The next closest sequence alignment is only 68% for a protein that is not expressed in the brain. The immunoreactive pattern observed in our histology reproduces work from mammalian models in which the epitope is available. Therefore, we are confident that our immunoreactive signal for vimentin is specific. We have added the in silico analysis in the manuscript on lines 535-538.

Comment 16 - Line 430: Was the GLM model used for testing all variables? Running a statistical model to explain Differentially expressed genes, photoperiod, and physiological variables together will give a more conclusive outcome to explain the photoperiod effect and seasonal state.

A similar comment was raised by Reviewer 2. We have conducted a WGCNA analyses to examine the relationship between photoperiod, physiological variables and DEG. (See Figures S3 & Table S6 and lines 166-182; 497-505).

Comment 17 - It is a bit unclear why the author used cherry-picking approach by talking about only a few genes that have been studied as key regulators of photoperiodic response in quail. What was the purpose of transcriptome? A better approach would have been to use a model to reduce the data (PCA) and explain the physiological response by regression against different PCs.

We agree with the Reviewer that other statistical approaches could be conducted, and other genes could be discussed. However, we focussed on the key regulators of the photoperiodic response in quail as these are the well characterised genes. It is important that our discussion focused on these transcripts as most do not conform to the predicted patterns of expression. We feel it is best that we keep the focus on these genes.

Comment 18 - TSHb result is inconsistent with past studies, where TSHb is the first responder gene on photoinduction. The author did not pay attention to explaining it further in the discussion.

We respectfully disagree with the Reviewer. Our results are consistent with past studies and show that TSHβ expression is a molecular marker of long day photoperiod. Our study does not examine photoinduction; which does not provide the ability to compare between our study and previous work (eg., Nakao et al., 2008; see doi: 10.1038/nature06738). We have revised the text in consideration of the concern raised by the Reviewer. The text now states ‘Previous reports established that TSHβ expression is significantly increased during the period of photoinducibility in quail (Nakao et al., 2008). Although the present study did not directly examine photoinduction, TSHβ expression was consistently elevated in long day photoperiod (i.e., 16L).’. (see lines 262-265).

Comment 19 - PRL result seems interesting and there could be more discussion in relation to the rise in PRL transcripts levels termination of breeding. Elaborating on PRL expression and breeding termination can add more information to the discussion.

This comment is not clear to us, and we would incorporate a clarified comment in a revised manuscript. The increased expression of prolactin does not occur during the termination of breeding. The increase in prolactin occurs during the vernal increase in photoperiod (ie 14L) but does not have a clear link with gonadal growth.

Comment 20 - Line 217-219: Based......respectively. Sounds like a big claim with less evidence.

We have removed the sentence from the discussion.

Comment 21 - Line 220-223: The .....Bird. The sentence is not clear about how this study would add to ecological studies. Need more clarity on the importance of such data.

The sentence was removed from the text.

Comment 22 - I think that it would be helpful to add a couple of caveats to provide more ecological context. First, the model is only based on males, and responses in females could be different.

We agree with the Reviewer there are undoubtedly sex differences in timing seasonal biology. However, the photoperiodic response (growth and regression) is similar in both males and females. Sex differences exist in response to supplementary environmental cues (e.g., temperature). Males were used in these studies as the gonadal response to changes in photoperiod manipulations are much larger compared to ovarian changes in females. The focus on males allows for fewer animals to be used in the experiments and greater statistical power. To address the Reviewers concern, we have added a paragraph in the discussion that describes the similarity in photoperiodic responses in males and females, and the importance of supplementary cues for full reproductive development in female birds. We also provide a couple sentences in the methods that describe the justification for only males in the present study. See lines (Methods 352-355; Discussion 312-330; and 334-339).

Comment 23 - Last, I wondered if it would be possible to add an ecological context for the frequent change in the photoperiod schedule and not take account of the endogenous annual response. Would the procedure simulate a similar kind of underlined molecular response for a bird under natural conditions responding to changing daylight cycles on an annual time frame?

The discussion was considerably revised to address the ecological relevance of the study, and findings. We have added a sentence at the beginning of the discussion to highlight that the laboratory-based approach and photoperiodic manipulations reliable replicate previous findings using semi-natural conditions (Robinson and Follett, 1982) (See lines 248-250). We have already reduced the focus on the endogenous annual response.

**Reviewer #2:**
Comment 1 - The writing is very terse and could benefit from a more narrating style, which would make it a lot easier for the reader to get through some of the very data-heavy text. Breaking up the Results with subheadings would also be helpful.

We appreciate the suggestion to add subheadings to the Results. We added 3 descriptive headings for each other studies conducted in the manuscript. We feel the added revision (e.g., ecological) has improved the narrative and made the manuscript accessible to the wider readership.

Comment 2 - The transcriptome analyses could be developed a bit more. First, using the limma package would allow the authors to apply a more complete model to the DEG analyses, which would likely be superior to EdgeR. Second, the authors may want to consider WGCNA or a similar approach to discover gene co-expression modules, and then examine whether any of the resulting module eigengenes co-vary with any morphological or physiological measures and/or vary rhythmically.

This is an excellent suggestion, and the new analyses was incorporated into the revised manuscript. Using the Langfelder and Horvath 2008 WCGNA package we conducted module-trait analyses to examine co-variation in our findings. These data are presented in Figure S# and lines 476-484. We agree that other DEG analyses would be useful; our main objectives was to use BioDare2.0 to identify rhythmic transcription in the seasonal transcriptomes. EdgR provides an excellent approach to identify transcripts and commonly used.

Comment 3 - In the Data and code availability statement (lines 226ff) the authors state that "all raw data are available in Extended data Table 1." However, they should be submitted to the GEO database or a similar public repository along with all relevant metadata. Also, and maybe I overlooked this, I did not see anywhere that the "R code used in Study 1 is freely available" (I was not sure what "the methods reference list" was supposed to refer to). Instead of stating that "the full R code used is available upon request" I suggest making all scripts available via GitHub or Dataverse, along with all non-omics data. The advantage of the latter platform is that a citable DOI is assigned to each upload.

The data are now available in the GEO database and can be accessed see GSE241775. We have added this information to the text. The R code is now provided as a Table S11 so that the reader can directly access the script.

Comment 4 - Line 191: Delete the extra "that"

We thank the Reviewer for identifying the oversight. We have revised the text accordingly.

Comment 5 - Line 24f: What does "pseudo-randomly" mean? Maybe "haphazardly" would be more appropriate here?

The term pseudo-randomly is used to describe the organized manner in which subjects are assigned to each treatment group. The aim is to ensure that a particular physiological variable, such as body mass, is evenly distributed across treatment groups. (Note although the term derived from the field of psychology). The aim is to reduce bias in the experiment due to an initial bias established when assigning treatment group. We are reluctant to replace pseudorandomly with haphazardly as the latter does not imply a logical organization. We have added text to help clarify the reason. The text now state: At the end of each photoperiodic treatment a subset of quail (n=12) body mass was used as a measure to pseudo randomly select birds for tissue collection and served to reduce the potential for unintentional bias.

Comment 6 - Figure 1e,j: The text indicates that 398 and 130 genes were "rhythmically expressed" in the MBH and pituitary, respectively, but considerably fewer genes are shown in the heatmaps in Figure 1e,j. How were these genes selected, and what was the rationale for doing so? Also, some autumnal and vernal expression patterns show some strong similarities (e.g., 16a and 16v in the MBH), which could be discussed. Consider showing the two heatmaps with the columns also hierarchically clustered in a supplementary figure.

We agree with the Reviewer that the full heatmap for the transcripts should be provided. The heat maps in Figure 1 are based on the transcripts with the most significant change; and were selected to provide a graphical representation that would be easily digested by the wide readership. We have created a new figure (ie. Fig. S1) that provides all the transcripts in heat maps for both the MBH and pituitary gland.

**Reviewer #3:**
Comment 1 I do not have too much to add to this section of my review. Broadly speaking, I would suggest that the authors address some of the concerns I highlight above, and integrate their thoughts into the paper more than they currently do. I think this is particularly important with respect to the limitations of many of the bioinformatic analyses.

We thank the reviewer for their input and time assessing the manuscript. We have revised the manuscript in many sections incorporating the suggestions by Reviewer 3 above, and Reviewers 1 and 2.

Comment 2 Some of the methods are also a little scant. For example, the qPCR analyses are not described in sufficient detail to replicate the study. What are the efficiencies? Were samples run in duplicate? What was the housekeeping control gene used? Was there only one, or were multiple housekeeping genes used?

We apologise for the oversight, the absence of information was a mistake that missed our previous early revisions. The revised manuscript includes all the requested information. Line 333 states that all samples were run in duplicate. The efficiency for each transcript was within the MIQE guidelines (indicated on line 342) and were within the 0.7 to 1.0 range. Actin and glyceraldehyde 3-phosphate dehydrogenase were used as the reference transcripts. The most stable reference transcript was used to calculate fold change in target gene expression (lines 343-345).